# The Bacterial Meningitis Epidemic in Banalia in the Democratic Republic of Congo in 2021

**DOI:** 10.3390/vaccines12050461

**Published:** 2024-04-25

**Authors:** Andre Arsene Bita Fouda, Anderson Latt, Abdoulaye Sinayoko, Franck Fortune Roland Mboussou, Lorenzo Pezzoli, Katya Fernandez, Clement Lingani, Berthe Miwanda, Dorothée Bulemfu, Francis Baelongandi, Patrick Mbenga Likita, Marie-José Kikoo Bora, Marcel Sabiti, Gervais Leon Folefack Tengomo, Eugène Kabambi Kabangu, Guy Kalambayi Kabamba, Issifou Alassani, Muhamed-Kheir Taha, Ado Mpia Bwaka, Charles Shey Wiysonge, Benido Impouma

**Affiliations:** 1World Health Organization Regional Office for Africa, Brazzaville P.O. Box 06, Congo; bwakaa@who.int (A.M.B.); sheyc@who.int (C.S.W.); impoumab@who.int (B.I.); 2World Health Organization Emergencies Hub, Dakar P.O. Box 36, Senegal; latta@who.int; 3World Health Organization Country Office Kinshasa, DRC, Kinshasa P.O. Box 06, Congo; sinayokoa@who.int (A.S.); folefacktengomog@who.int (G.L.F.T.); kabambie@who.int (E.K.K.); kalambayikabamba@who.int (G.K.K.); 4World Health Organization Inter-Country Support Central Africa, Libreville P.O. Box 820, Gabon; mboussouf@who.int; 5World Health Organization (WHO), 1202 Geneva, Switzerland; pezzolil@who.int (L.P.); fernandezk@who.int (K.F.); 6World Health Organization Inter-Country Support West Africa, Ouagadougou 03 BP 7019, Burkina Faso; linganic@who.int; 7Institut National de Recherche Biomédicale, DRC, Kinshasa P.O. Box 1192, Congo; 8Ministry of Public Health Hygiene and Prevention, DRC, Kinshasa P.O. Box 1192, Congo; dobulemfu@yahoo.fr (D.B.); francisbaelongandi@gmail.com (F.B.); likitamb1@gmail.com (P.M.L.); kikooboram@who.int (M.-J.K.B.); dr.marcelsabiti@yahoo.fr (M.S.); 9World Health Organization Country Office, Lome P.O. Box 1504, Togo; alassanii@who.int; 10Institute Pasteur of Paris, 75015 Paris, France; muhamed-kheir.taha@pasteur.fr

**Keywords:** meningitis, banalia, meningococcal vaccine, Democratic Republic of Congo

## Abstract

Background: The Banalia health zone in the Democratic Republic of Congo reported a meningitis epidemic in 2021 that evolved outside the epidemic season. We assessed the effects of the meningitis epidemic response. Methods: The standard case definition was used to identify cases. Care was provided to 2651 in-patients, with 8% of them laboratory tested, and reactive vaccination was conducted. To assess the effects of reactive vaccination and treatment with ceftriaxone, a statistical analysis was performed. Results: Overall, 2662 suspected cases of meningitis with 205 deaths were reported. The highest number of cases occurred in the 30–39 years age group (927; 38.5%). Ceftriaxone contributed to preventing deaths with a case fatality rate that decreased from 70.4% before to 7.7% after ceftriaxone was introduced (*p* = 0.001). *Neisseria meningitidis* W was isolated, accounting for 47/57 (82%), of which 92% of the strains belonged to the clonal complex 11. Reactive vaccination of individuals in Banalia aged 1–19 years with a meningococcal multivalent conjugate (ACWY) vaccine (Menactra^®^) coverage of 104.6% resulted in an 82% decline in suspected meningitis cases (incidence rate ratio, 0.18; 95% confidence interval, 0.02–0.80; *p* = 0.041). Conclusion: Despite late detection (two months) and reactive vaccination four months after crossing the epidemic threshold, interventions implemented in Banalia contributed to the control of the epidemic.

## 1. Introduction

Bacterial meningitis is a major public health problem in the African meningitis belt [1,2,3,4,5,6,7]. Seasonal meningitis outbreaks occur annually in this part of the world mostly from epidemiological weeks (w) 1 to 26 (January–June) [8,9]. Despite significant progress in combating meningitis over the past 20 years, bacterial meningitis epidemics remain a significant global public health challenge, with over 1.2 million cases and 300,000 deaths occurring annually [7]. The incidence and case fatality rates of bacterial meningitis vary by region, country, pathogen, and age group [1]. In 2019, approximately 2.5 million cases and 236,000 deaths due to meningitis were reported worldwide [7,10]. Before 2010, *Neisseria meningitidis* (*N. meningitidis*) A was the leading cause of meningitis in the African meningitis belt, accounting for almost 90% of the epidemics [7]. The introduction of meningitis A conjugate vaccine (MenAfriVac^®^) in the African meningitis belt since 2010 resulted in a significant reduction in the incidence of *N. meningitidis* A cases and a change in the bacterial profile of meningitis, with a predominance of *N. meningitidis* C, W, X, and *Streptococcus pneumoniae* (*S. pneumoniae*) [8,10]. Since 2010, countries in the meningitis belt including Burkina Faso (BFA), Ghana, Nigeria, Niger, and Togo have reported meningitis epidemics caused by *N. meningitidis* C and W [11,12,13,14,15,16,17,18,19,20,21]. The clonal complex (CC)11 has expanded throughout the meningitis belt [18]. Although the detection and laboratory confirmation of pathogens causing meningitis epidemics is challenging in these countries, reactive vaccination is vital for controlling them. Documenting lessons learned from responses helps identify bottlenecks and best practices and improves the quality of preparedness, detection, and response over time [11,12,13,14,15,16,17,18,19,20].

The Democratic Republic of Congo (DRC) is in the African meningitis belt. The country last reported a meningitis epidemic in 2009, in the city of Kisangani, with 214 meningitis suspected cases with a case fatality rate (CFR) of 8% [21,22]. In May 2016, MenAfriVac^®^ was introduced in four provinces, with over 18 million individuals aged 1–29 years vaccinated, 1.7 million of whom resided in Tshopo province [23,24]. In early July 2021, the Banalia health zone in the Tshopo province, located in the north-eastern DRC, notified several suspected cases and deaths due to meningitis through the alert system of the meningitis enhanced surveillance network of the African meningitis belt in which DRC was included. On 6 September 2021, *N. meningitidis* W was identified as the cause by real-time polymerase chain reaction (PCR) at the Institute Pasteur of Paris, France [21,22,25]. We report on the challenges and best practices in implementing outbreak response interventions to assess their effects on the epidemiological evolution of the bacterial meningitis epidemic in Banalia.

## 2. Methods

### 2.1. Setting

This cross-sectional study was retrospectively conducted from July to December 2021 in the Banalia Health Zone, which comprises 20 health areas with 171,001 inhabitants. As the World Health Organization (WHO) recommends dividing areas with over 100,000 inhabitants into smaller intervention areas, two subhealth zones were identified, the left and right banks, with 88,311 and 82,690 inhabitants, respectively. Outbreak response interventions were implemented by the government with support from partners, including the WHO [25,26,27].

### 2.2. Meningitis Surveillance

The DRC has the following two complementary meningitis surveillance systems: enhanced surveillance, which was introduced in 2003 in the six provinces in the meningitis belt (Bas Uele, Haut Uele, Ituri, Nord Kivu, Sud Kivu, and Tshopo), and pediatric bacterial meningitis sentinel surveillance, which was implemented in 2009 in three sites (Kinshasa, Lubumbashi, and Kisangani). The meningitis-enhanced surveillance aims to detect outbreaks. A suspected case of meningitis was defined by fever, neck stiffness, and one or more neurological signs [28]. Enhanced and community-based surveillance was implemented by public health authorities during the epidemic, starting active case-finding on 17 September (w37), 2021 [27,29,30]. As per WHO guidance, subhealth zones were classified as crossing the alert or epidemic thresholds when 3 or 10 suspected cases per 100,000 inhabitants per week were recorded, respectively [27,28,29,30].

### 2.3. Laboratory Confirmation

Detection of bacterial pathogens is performed by culture or PCR in cerebrospinal fluid (CSF) specimens [28]. To confirm the diagnosis, lumbar punctures were performed in the health facilities of Banalia on some suspected cases of meningitis. First-level laboratories in Kisangani performed Gram staining and latex agglutination (Pastorex^®^). Aliquots of all CSF specimens were shipped to the National Institute of Biomedical Research of Kinshasa for culture and PCR. On 6 September 2021, the Pasteur Institute of Paris (IPP) confirmed that *N. meningitidis* W was the main cause of the epidemic. A total of 213 isolates of CSF specimens were shipped to the IPP and the Centers for Disease Control and Prevention (CDC) of the United States for quality control and testing for antimicrobial resistance using the minimum inhibitory concentration method and molecular genotyping. Molecular typing was directly performed on CSF samples or cultured isolates by PCR amplification followed by sequencing of several genes to perform multi-locus sequence typing and fine typing [31]. Molecular antimicrobial testing was used for predicting susceptibility to beta lactams by penA sequence analysis as previously described [32]. Antimicrobial susceptibility testing was performed as recommended by the European Monitoring Group on Meningococci [33]. Ebola virus disease and any heavy metal poisoning were excluded through biological and biochemical laboratory tests. Eight hundred suspected meningitis cases benefited from latex agglutination rapid diagnostic testing for malaria. One patient with respiratory symptoms was tested for coronavirus disease 2019 (COVID-19) using a rapid diagnostic test, and PCR was not performed for confirmation.

### 2.4. Public Health Response

The national health authorities provided support in responding to this outbreak in coordination with the WHO, United Nations Children’s Fund, Gavi, The Vaccine Alliance, International Coordinating Group (ICG) on Vaccine Provision, CDC of the United States, Doctors Without Borders (MSF), World Bank, and civil society. The response measures included the deployment of national and provincial rapid response teams that conducted investigations and organized response structures and mobile clinics to ensure appropriate case management with ceftriaxone administration, sample collection, and IPC measures to prevent COVID-19 comorbidity in the affected areas. To determine the epidemiological linkage, in-depth investigations at the community level were conducted. Alerts and active searches for contacts were established through community-based surveillance in the mining quarries and community. Patients who were victims of another traumatic event (mourning a loved one or having a serious illness) after recovery from meningitis were excluded from the study. To screen for depression, the HAD scale by Zigmond and Snaith was administered more than 1 month following meningitis management. Any case whose HAD score was over 10 was considered a case of depression. Risk communication activities were implemented. The local, provincial, and national coordination committees of health emergencies conduct regular meetings [27,34,35,36,37].

### 2.5. Data Collection and Statistical Analysis

From January to December 2021, data were retrospectively collected from health facility registers and reported in a line list. The incidence rates of suspected meningitis cases were calculated for each epidemiologic week of the two subhealth zones. To guide public health response, rates at both health and subhealth zone levels were compared with WHO-established thresholds [15,16]. Vaccine administrative coverage was estimated using the number of vaccinated individuals and the target population of 146,990 1–49-year-old individuals. Laboratory data were included in the analysis. Only 2444 of 2662 meningitis suspected cases were used for statistical analyses owing to missing data. All 213 (100%) CSF specimens were adequate. They were analyzed either by PCR (n = 114) or culture (n = 102), and 112 CSF samples were tested using both methods. To determine differences between the distribution of suspected cases among age groups, gender, and status of alive and dead, the Kruskal–Wallis rank sum test was employed. The Wilcoxon–Mann–Whitney test was used to assess the relationship between the number of cases that occurred before and after reactive vaccination, and the incidence rate ratio (IRR) was determined to be the correlation between the number of deaths that occurred before and after the introduction of ceftriaxone. The confidence interval used was 95% with a significance level of <0.05.

### 2.6. Ethics Considerations

During the study, respect for human beings was followed. The anonymity and confidentiality of patients were respected. Research and publication authorization from the Ministry of Health were obtained.

## 3. Results

### 3.1. Outbreak Detection, Investigation, and Spread

The meningitis epidemic in Banalia was detected during the 26th epidemiological week (w) of 2021 at the beginning of July and officially declared late on 7 September 2021 (w35). The first cases were recorded in the Panga Health area located 277 km north of Kisangani and in the Wabelo and rapid intervention mine quarries 4 km upstream and 6 km downstream of the Panga Health facility, respectively. From this health area, the epidemic spread to the nineteen other health areas in the Banalia health zone. The right bank subhealth zone crossed the epidemic threshold in w21 (11.7/100,000 inhabitants), whereas the left bank subhealth zone in w25 (10/100,000 inhabitants). The right bank subhealth zone was the most affected, with a cumulative attack rate of 3327 cases/100,000 population with 107 deaths and a 9.2% lethality rate (Table 1). This epidemic was unusual in that it evolved after the meningitis epidemic season (w1–26).

The end of the epidemic was officially declared in w50 (23 December 2021). The epidemic lasted 31 weeks with a peak in week 40 (4 October 2021) (Table 1). At w44 (1 November 2021), 2449 suspected and 213 confirmed meningitis cases with 205 deaths had been recorded, with a CFR of 7.7%.

The highest number of cases occurred in the 30–39 years age group 30–39 years (927/2409 [38.5%]), followed by those aged 15–29 years (647/2409 [26%]), ≥50 years (396/2409 [16%]), 5–14 years (251/2409 [10%]), and 0–59 months (188/2409 [7.7%]) (*p* < 0.001); of the 2,662 suspected cases, 253 (9.5%) had missing data. Males and females represented 1280/2447 (52.5%) and 1163/2447 (47.5%) suspected cases, respectively. However, no statistical difference was observed between males and females (*p* = 0.8). A total of 2457/2662 (92.3%) suspected cases were treated, and 205 cases died (CFR= 7.7%). The proportion of alive individuals was higher than that of deaths (Table 2).

### 3.2. Laboratory Confirmation

Of 2662 suspected cases, 213 (8%) CSF specimens were collected and tested. Of the 213 specimens, 57 (26.7%) were positive for bacterial meningitis pathogens by at least one confirmation method. Of the 57 specimens with positive confirmatory test results, 47 (82%), 4 (7%), 3 (5%), 2 (3%), and 1 (2%) were identified as *N. meningitidis* W, *N. meningitidis* C, *S. pneumoniae*, *Hemophilus influenzae b*, and *Hemophilus influenzae non-b*, respectively (Table 3). The molecular genotyping performed by IPP showed a W genome: P1.5.2. F1–1: clonal complex CC11 (92%), and the strains belonged to the Anglo–French–Hajj lineage. One suspected meningitis case that tested positive for COVID-19 using a rapid diagnostic test died. In contrast, 800/2662 (30%) suspected meningitis cases were tested using a rapid diagnostic test for malaria, and 77/800 (9.6%) were positive.

### 3.3. Case Management

From 7 June to 4 August 2021, the treatment protocol included ampicillin and gentamycin with a cumulative CFR of 70.4%. Subsequently, on 5 August 2021, ceftriaxone was introduced, replacing ampicillin and gentamicin, and the cumulative CFR decreased to 7.7% at the end of the epidemic, with 205 deaths of 2662 suspected cases (Figure 1).

A correlation was noted between the number of deaths before and after the introduction of ceftriaxone into the treatment protocol replacing ampicillin gentamicin (*p* = 0001). Statistical analysis showed a high degree of correlation between the reactive vaccination campaign with Menactra^®^ and the occurrence of cases (*p* = 0.001) (Table 4).

A correlation was also noted between ceftriaxone administration and lethality (*p* = 0.013), and ceftriaxone administration markedly reduced lethality (IRR, 12.5; 95% confidence interval [CI], 1.48–114).

### 3.4. Psychological Support

Psychological support was provided to patients, their families, and the community, and 5442 individuals benefited from psychoeducation. Psychological support was provided to 761 patients, 391 (51.4%) of whom experienced psychological conditions (sadness, anxiety, insomnia, and erectile dysfunction). A total of 321 health workers and community agents who were assigned to the epidemic response and 182 patients before lumbar puncture also benefited from psychological support. Conversely, 275 children with malnutrition and orphans with meningitis as well as pregnant and breastfeeding women benefited from specific nutritional support. Moreover, 5504 individuals were sensitized to sexual exploitation and abuse prevention.

### 3.5. Infection Prevention Control

IPC was implemented from w31 to w50 in 2021. IPC encompassed cleaning and disinfection of latrines and meningitis treatment centres, chlorination of handwashing points, and promotion of wearing masks in Banalia public areas and health facilities.

### 3.6. Reactive Immunization

This study showed that after reactive vaccination, an 82% decline in the number of meningitis-suspected cases was observed, which contributed to ending the epidemic (IRR, 0.18; 95% CI, 0.02–0.80; *p* = 0.041) (Table 5).

In addition, reactive vaccination was conducted from 9–16 October 2021. The target of the reactive vaccination campaign was 146,990 inhabitants of Banalia aged 1–49 years. A total of 15,388 inhabitants of Banalia, miners (coming from other cities) and travellers who went to the towns upstream of the Banalia river were vaccinated, and the coverage was of were vaccinated with a coverage rate of 104.6% (Figure 2).

### 3.7. Coordination, Monitoring, and Evaluation

Technical coordination meetings of the response were organized daily under the leadership of the Director of Health of Tshopo province. Other coordination platforms were established. Meetings of the local health emergency management committee of the Banalia health zone and coordination meetings of the three WHO levels (headquarters, region, and country), which were chaired by the WHO Regional Office for Africa, with the participation of partners, were held. Monitoring of activities and supportive supervision were conducted in the field by the Ministry of Health’s staff from the provincial and national levels with support from the WHO and partners. An after-action review of the meningitis epidemic response was organized from 25–27 March 2022 in Kisangani and observed that significant achievements have been made in terms of rational use of resources, involvement, and accountability of local actors. However, implementation of a clear mechanism for managing the epidemic at the operational level especially following standard operating procedures of detection and response to the meningitis epidemic recommended by the WHO was lacking. In fact, the utilization of the epidemic threshold in week 25 would have shown that Banalia was in epidemic and CSF samples of suspected cases would have been collected and tested to confirm the epidemic earlier. Again, the appropriate treatment protocol with ceftriaxone could have been used at the beginning to avoid the high lethality by using ampicillin and gentamicin. Knowing the cause of this epidemic (N meningitidis W) earlier would have helped to request ICG on-time vaccines for the reactive vaccination campaign.

### 3.8. Risk Communication and Community Engagement

The 186 community leaders from the Lukelo, Tshololo, Ste Elisabeth, and Bogbama health areas were informed and sensitized through communication on the risks of the meningitis epidemic, its possibility of worsening and the attitude and practices of the community to slow down and stop this epidemic. A total of 275 individuals from 35 households in the Mangala health area were also reached. Community leaders were involved in community-based surveillance and reported alerts to treatment centres. A local community radio station called “Canal Aruwimi Radio” contributed to sensitization programs based on meningitis. Key messages on collective and individual protective measures and spots following a frequency of three broadcasts per day were delivered.

## 4. Discussion

With 2662 suspected cases and 205 deaths, the Banalia meningitis epidemic is one of the largest epidemics caused by *N. meningitidis* W reported in the meningitis belt countries over the last decade. Other countries that declared large epidemics of *N. meningitidis* W included BFA, Niger, and Togo [12,14,15]. The WHO recommends early detection of meningitis epidemics [28,29,30]. However, the alert and epidemic thresholds were not applied at the beginning of the epidemic. This explains why this epidemic was detected 5 weeks later when the right bank subhealth zone had already crossed the epidemic threshold. From w21 to w30, the CFR was very high at 70.4%, probably because of ampicillin and gentamycin administration, which are not recommended by the WHO. When ceftriaxone was introduced, the CFR markedly decreased from 70.4% to 7.7% in w47. A CFR of ≥10% is considered high [28]. Furthermore, Togo and BFA reported a CFR of 10% [12,13,17]. The age group most affected by meningococcal meningitis is typically the 1–29-year-old age group [30]. In Banalia, meningitis incidence was very high among adults, such as in Togo in 2016 [12], whereas children were the most affected in Niger and BFA [14,15]. Males (1280/2447 [52.4%]) were more affected than females (1163/2447 [47.6%]). This result is similar to epidemics reported in BFA, Niger, and Togo [12,14,15].

Regarding case management, the definition of a suspected meningitis case, which was the one recommended by the WHO, allowed the correct identification of patients for treatment [28]. At the start of the alert phase, a combination of ampicillin and gentamicin was used for treatment, which was inappropriate because the WHO recommends ceftriaxone [28,30]. The chemoprophylaxis regimen deviated from the WHO recommendations [28,30]. Health authorities of the Ministry of Health disagreed and justified that it was to avoid antimicrobial resistance to ciprofloxacin. Expectedly, based on experiences from other countries including BFA, Togo, Niger, and Benin, ceftriaxone administration markedly decreased the CFR [12,13,17,18]. Health authorities ensured that aftercare was in place, offering psychological support to patients and health workers. Finally, to reduce the risk of comorbidity (e.g., COVID-19) and further transmission of meningitis within the healthcare setting, IPC activities were implemented [19,20,21].

The number of CSF specimens collected (8%) was very low, and at least 50% should be collected [28]. Despite the low confirmation rate, we identified CC11 as the main cause of this epidemic circulating in the African meningitis belt over the last 20 years [12,13,17,18]. The strains belong to the Anglo–French–Hajj lineage, which has occurred in different sublineages across Africa [18].

Reactive vaccination seems to have contributed to the dramatic decline in cases, as shown in other countries (BFA, Niger, Benin, and Brazil) [12,14,15,34]. However, the campaign, which started in w40, was very late, almost 20 weeks after the start of the epidemic. WHO recommends vaccination within 4 weeks following meningitis epidemic detection [18,20]. Unfortunately, delays in initiating reactive vaccination frequently occur in countries. Recent reactive campaigns in BFA, Niger, and Togo have also reported delays [12,13,17]. However, in all these cases, vaccination was initiated <12 weeks following epidemic detection.

Coordination was initially weak and scaled up progressively following the official declaration of the epidemic [19,20,21]. Partners contributed to strengthening the epidemic response. The after-action review supported by the WHO contributed to evaluating preparedness and response to the epidemic and identified as lessons learned the need for earlier detection and faster reactive vaccination [19,20,21].

This study had some limitations. First, 253/2662 (9.5%) and 215/2662 (8%) missing data on age and gender, respectively, were observed. Therefore, to estimate the proportions of age and gender, 2409 and 2447 were designated as the denominator, respectively. This situation can be explained by the poor quality of data reporting in case investigation forms in a few health facilities in Banalia. Second, the proportion of CSF specimens tested was very low (213/2662 [8%] cases) because of a lack of lumbar puncture kits and only a few health workers capable of performing lumbar puncture. The WHO recommends at least 50% of CSF collection among meningitis suspected cases [28].

In conclusion, despite the late detection of the bacterial meningitis outbreak in Banalia, adequate case management including the best practice of setting up psychosocial aftercare and conducting a reactive vaccination campaign made it possible to reduce lethality, stop the epidemic, and reduce its negative effects on the affected population. We recommend allocating resources to reinforce laboratory capacity for quicker detection and confirmation of meningitis cases and applying the alert and epidemic thresholds as quickly as possible to guide a timely response.

## Figures and Tables

**Figure 1 vaccines-12-00461-f001:**
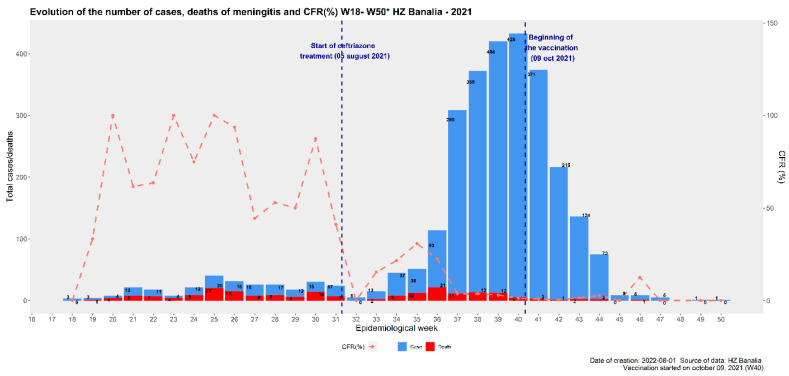
Evolution of meningitis suspected cases, deaths, and case fatality rates in Banalia, Democratic Republic of Congo from 7 June to 27 November 2021.

**Figure 2 vaccines-12-00461-f002:**
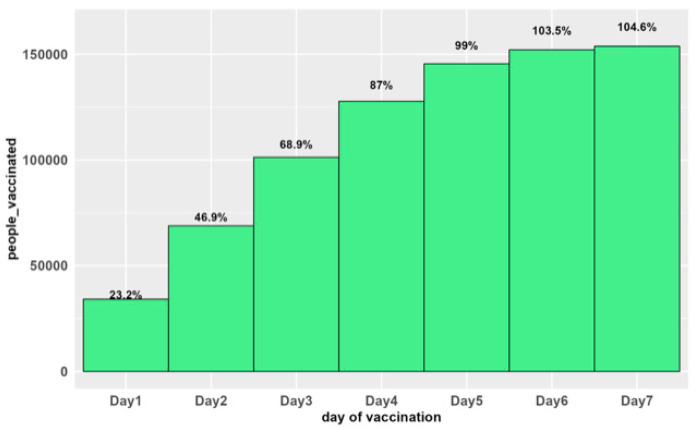
Daily evolution of Menactra^®^ vaccination coverage in Banalia, Democratic Republic of Congo from 9–16 October 2021. Dash line—CRF (%), blue column—Case, red column—Death.

**Table 1 vaccines-12-00461-t001:** Characteristics of the meningitis epidemic in right bank and left bank sub-health zones, Tshopo, Banalia, Democratic Republic of Congo from 7 June to 27 November 2021.

Sub-Health Zones	Population	Cumulative Suspected Cases	Cumulative Attack Rate (per 100,000 Inhabitants)	Weeks in Which Epidemic Threshold Was Crossed (Number of Suspected Cases)	Cumulative Deaths	Case Fatality Ratio (%)
Right bank	79,319	1844	2326.1	w21 (13), w22 (8), w24 (10), w25 (20), w26 (16), w27 (18), w28 (17), w30 (14), w31 (10), w33 (13), w34 (34), w35 (25), w36 (61), w37 (217), w38 (223), w39 (258), w40 (312), w41 (253), w42 (125), w43 (108), w44 (56)	170/2326.1	9.2%
Left bank	88,286	818	926.5	w35 (14), w36 (32), w37 (81), w38 (136), w39 (150), w40 (116), w41 (118), w42 (90), w43 (26), w44 (17)	35/926.5	4.3%
Banalia health zone	167,605	2662	1588.9	W 21–44	205/2662	7.7%

**Table 2 vaccines-12-00461-t002:** Distribution of age groups, sex, alive and deaths in Banalia in Banalia health zone, Tshopo, Democratic Republic of Congo from 7 June to 27 November 2021.

Variables	Number (n^1^)	Meningitis Suspected Cases (n^2^/N^3^)	*p*-Value
**Age group**			
0–59 month		188/2409 (7.8%)	
5–14 year		251/2409 (10.5%)	
15–29 year		647/2409 (26.8%)	<0.001 *
30–49 year		927/2409 (38.5%)	
50 year and above		396/2409 (14.4%)	
Missing data		253/2662 (9.5%)	
**Gender**			
Female		1163/2447 (47.6%)	0.8 **
Male		1284/2447 (52.4%)	
Missing data		215/2662 (8%)	
**Status**			
Alive		2457/2662 (96.3%)	<0.001 *
Death		205/2662 (7.7%)	

n^1:^ Number; n^2^: number case N^3^: total number; ***** Kruskal–Wallis’s rank sum test, ****** Wilcoxon rank sum test.

**Table 3 vaccines-12-00461-t003:** Meningitis pathogens isolated confirmed during the meningitis epidemic in Banalia health zone, Tshopo, Democratic Republic of Congo from 7 June to 27 November 2021.

Pathogens Confirmed	Meningitis Pathogens n^1^/N^2^ (%)
*N. Meningitidis W*	47/57 (82.5%)
*N. Meningitidis C*	4/57 (7%)
*S. pneumoniae*	3/57 (5%)
*Haemophilus influenzae b*	2/57(3,5%)
*Haemophilus influenzae non-b*	1/57 (2%)
Total	57/57 (100%)

n^1^: Number of pathogens positive, N^2^: 57 CSF collected and tested positive.

**Table 4 vaccines-12-00461-t004:** Relationship between the number of cases and deaths before and after reactive vaccination and ceftriaxone introduction.

	Number of Cases		*p*-Value	Number of Deaths		*p*-Value
	Before Reactive Vaccination w 21–40	After Reactive Vaccination w 41–47		Before Ceftriaxone Introduction w 21–31	After Ceftriaxone Introduction w 32–47	
Cases and deaths before and after interventions	1745	908	*p* = 0.001 ^1^	111 deaths	94 deaths	*p* = 0.001 ^1^

^1^ Wilcoxon rank sum test.

**Table 5 vaccines-12-00461-t005:** Correlation between meningitis incidence before and after reactive vaccination and number of related deaths before and after ceftriaxone introduction in the treatment protocol.

Characteristic	IRR ^1^	95% CI ^2^	*p*-Value
Reactive vaccination campaign			
Before (suspected cases)	—	—	
After (suspected cases)	0.18	0.02, 0.80	0.041
Ceftriaxone administration			
Before (deaths)	—	—	
After (deaths)	12.5	1.48, 114	0.013

^1^ IRR = Incidence Rate Ratio, ^2^ CI = Confidence Interval.

## Data Availability

The data can be shared up on request.

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
