# Peer review of "The Bacterial Meningitis Epidemic in Banalia in the Democratic Republic of Congo in 2021"

_vaccines, 2024, doi:10.3390/vaccines12050461_

Round 1
Reviewer 1 Report
Comments and Suggestions for Authors
This report of a large outbreak of N meningitidis W, later in the year than is common for such outbreaks, is well done and important to document in the literature, to warn others and work toward quality improvement in such efforts.
There are several small areas that will benefit from clarification, some numbers require a second check.
1. The time from first case to recognition to reporting to response is one of your major findings and should be specifically noted in the abstract.
2. Some of the same information is included in the introduction, methods, and results. For example, the week the first case occurred, the week reponse began. It is reasonable to include brief mention in the introduction, as you do; but remove such details from Methods, and in Methods point out that you examine and report on response times, etc. For example, much of the information in sections 2.4 and 2.5 are more clearly explained in Results and can be left out of the Methods section.
3. Line 95: please clarify why there are different epi thresholds for response in two areas so close to each other.
4. Beginning on line 48 and again in results, you touch on psychological and other responses, beyond epidemic surveillance and control. these details might be better reported in a separate paper, but if you are keeping them here, more explanation is needed. For example, is depression a known complication of N meningitidis? Or is it a result of the scarey fast moving outbreak. Also this occurred at height of worldwide Covid pandemic, was that contributing to depression? If you keep this information you need a transition or new header to let the reader know this is additional response..
5. Section 3.2: Laboratory Confirmation: Of the 213 CSF specimens, only 26.7% were positive for bacterial pathogens. Did the other patients not have bacterial meningitis? Why do you say you had 2662 suspected cases, were some ruled out? Were there false negatives in the testing? Or did 74% of your suspect cases have some illness other than N meningitidis?
6. Section 3.3 including the two charts is very well done However please make sure you are correct that all 2662 were part of the same outbreak (see comment 5). Lines 246-248: You cannot cover more than 100% of your target. Did people not in your target travel in for vaccine? Please clarify.
7. Paragraph starting line 249: as noted above, justification is needed, perhaps in your introduction, for including this information.
8. Lines 272-274: please clarify what WHO recommendations and which were not followed.
9. Line 277 and 281: What is meant by "sensitization:? Do you mean training in risk communication?
10. Did you conclude that population wide use of menactra is now under consideration?
Minor comments / word edits
LInes 31-32: you need more in this sentence, e.g. after ceftriaxone was introduced.
Line 68: Please say who DRC notified.
Line 89: Do you mean Kinshasa or is there another place called Kinsasha?
Line 182: consider changing "approved" to "obtained".
LIne 185: this says epidemic detected in week 26; elsewhere you say week 21.
Lines 196-197: perhaps make this two sentences: ...end ...declared week 50. The epidemic had lasted 31 weeks with a peak in week 40 (Better to give the dates always when you say the week)
Lines 209-210: Omit last sentence. Just say CFR.
This is an important report, thank you for doing it.
Comments on the Quality of English Language
I have noted some possible word changes above, e.g. the word Sensitization as used is not clear to me.
But overall the paper is well written.
Author Response
Thank you so much for the time spent to review our paper. all the suggestions made are relevant and we accepted

Reviewer 2 Report
Comments and Suggestions for Authors
The manuscript is of significant interest to general and specific readers.
Did the authors have access to potential clinical manifestations of the meningitis epidemic.
Specifically, was rash described?
Were there other manifestations such as septic arthritis or peri-joint infection (tenosynovitis)?
Author Response
Thank you very much for your relevant comments.
Reviewer 3 Report
Comments and Suggestions for Authors
Thank you for your submission.
It is an interesting manuscript.
The statement, used in several places, “Individuals 30 aged 30–39 years were the most affected, with 927/2,409 (38.5%) cases” is not correct. To describe the age group most affected, we would need to know the infection rate per 1000 in each group of the whole population. Instead, the authors should state “The highest number of cases occurred in the 30-39 years age group (922,40; 38.5%)”. Alternatively, infection rates should be calculated for each age group of the whole population.
This statement and the statistical test does not make much sense “The proportion of alive individuals was higher than that of deaths with statistical 209 difference (p < 0.001) (Table 2).” It should be just a statement without any statistical test (which is meaningless in this context).
Are there any data on sequelae in the survivors of the infection?
There is repetition in the discussion about the appropriate selection of antibiotics.
Were Tables 4 and 5 included? I could not see them. They are probably unnecessary anyway (and would make the manuscript overly long) as Figure 1 nicely display the key findings.
Comments on the Quality of English LanguageOnly minor issues found.
Author Response
Thank you so much for your relevant suggestion. We agreed. Concerning tables 4 and 5 they were added.
